# Evaluating the CRP Interactome: Insights into Possible Novel Roles in Cellular Signaling and Tumorigenicity

**DOI:** 10.3390/cimb47121003

**Published:** 2025-11-28

**Authors:** Alison Gerhardt, Dlen Nanakaliy, Harnish D. Shah, Syed Sarfaraz, Charlotte Madigan, Makenna Downing, Robert P. Elliott, Rafid Dipto, Bella Vrapciu, Joseph W. Noetzel, Jessica Armas, Ayesha Ahmed, Marc Potempa, Margaret E. Olson, Lawrence A. Potempa, Peter C. Hart

**Affiliations:** 1College of Science, Health and Pharmacy, Roosevelt University, Schaumburg, IL 60173, USA; agerhardt@mail.roosevelt.edu (A.G.); relliott02@mail.roosevelt.edu (R.P.E.); rdipto@mail.roosevelt.edu (R.D.); jnoetzel@mail.roosevelt.edu (J.W.N.); jessicaarmas12345@gmail.com (J.A.); aahmed36@mail.roosevelt.edu (A.A.); lpotempa01@roosevelt.edu (L.A.P.); 2Acphazin, Inc., Deerfield, IL 60015, USA

**Keywords:** C-reactive protein, CRP, interactome, protein-protein interactions, glycoproteins, glycosaminoglycan biosynthesis, extracellular matrix, tumor microenvironment

## Abstract

C-reactive protein (CRP) is a well-known acute phase reactant and putative biomarker for advancing and chronically established inflammation. Its biological activity across its multiple isoforms plays various roles in the initiation, potentiation, and resolution of inflammation. Its molecular signaling within the tissue microenvironment regulates cell–cell communication across cell types (e.g., epithelial cells, endothelial cells, fibroblasts, adipocytes, and immune cells) and affects the development of conditions such as cancer that are subject, at least in part, to inflammatory signaling. Considering the dynamic nature of CRP in modulating disease progression, and the growing evidence of the context-dependent direct molecular activity of CRP on regulating intra- and inter-cellular signaling, it is critical to further understand how this integral molecule alters cell signaling pathways. Although the ability of CRP to directly interact with some extracellular matrix proteins involved with inflammation and disease has been reported as early as the mid-1980s, recent advances in unbiased proteomics have revealed a broader interactome of protein–protein interactions (PPIs) involving CRP. The present study evaluates the CRP PPIs identified to date and explores the potential novel regulatory capacity of CRP on multiple key cellular functions in metabolism and cell–cell signaling, offering an updated framework of the possible biological activities of CRP relevant to tumorigenic processes.

## 1. Introduction

C-reactive protein (CRP) is one of the most well-known acute phase reactants, traditionally serving as a diagnostic biomarker for inflammation and disease progression. High expression of CRP has been associated with worse outcomes in many inflammatory diseases, including rheumatoid arthritis, diabetes, and cardiovascular disease [1,2,3,4,5]. High CRP levels are also often prognostic of cancer incidence and advancement [6], with evidence showing CRP can facilitate tumor growth, metastatic behavior, chemoresistance, and immune evasion in various contexts [7,8,9,10,11]. However, other data suggests CRP has tumoricidal and anti-proliferative activities, highlighting a critical need to investigate the role of CRP in the tumor microenvironment (TME) so that its activities may be more advantageously regulated [12].

There are three known isoforms of CRP whose distinct interactions and effects likely underlie the aforementioned variability in CRP-mediated effects. The predominant form of CRP is that of a pentamaric ring with five-fold symmetry (pCRP). This native form is inert or weakly anti-inflammatory. However, ligand binding and membrane association at sites of tissue injury initiate conformational changes into the generally pro-inflammatory isoforms of pCRP* (a modified pentamer) and mCRP (a monomeric version formed after pCRP dissociation). The conventional ligands recognized by pCRP and pCRP* are phosphocholine-containing carbohydrates in lipid rafts and effector molecules of the complement system [13,14,15,16,17]. Monomeric CRP retains the ability to bind phosphocholine and complement, but its signature ligand is alternatively considered to be cholesterol within cellular membranes. However, these ligands are far from CRP’s only interacting partners. Several alternative interacting partners have been identified, and recently published proteomic data has even further extended that list significantly. In this review, we evaluate both the established and novel protein–protein interactions (PPIs) for CRP with an emphasis on discerning the network of interactions that underly CRP’s role in modulating tumor development, progression, and treatment resistance.

## 2. The Updated C-Reactive Protein Interactome

In addition to its conventional effector ligands of the complement system [18,19], reports have identified the ability of CRP to directly interact with extracellular matrix (ECM) proteins, such as fibronectin [15,18,20,21], laminin [14], and collagen IV [18]. Recently, broad unbiased proteomic analysis has complemented the existing network of protein–protein interactions (PPIs) of a myriad of proteins, which included CRP. Briefly, Huttlin and colleagues had utilized the open reading frame (ORF) plasmid set “ORFeome” [22], using FLAG-tagged genes transduced in cells in vitro, to allow the immunoprecipitation of target (“bait”) protein-bound complexes measured through affinity purification mass spectroscopy (AP-MS) to identify protein–protein interactions [23]. This study provided a novel interface for assessing the biophysical interactions of these ORFeome-based complexes (BioPlex Explorer) in the cell lines HEK293T (an epithelial-like cell derived from embryonic kidney) and HCT116 (a colorectal carcinoma tumor epithelial cell). We utilized the BioPlex Explorer database to query proteins immunoprecipitated by CRP pulldown, as identified by AP-MS, which revealed 57 novel interacting proteins in the CRP network (Figure 1A,B). There were 19 PPIs immunoprecipitated and identified in both the HEK293T and HCT116 cell-based AP-MS assays, with an additional 24 and 14 unique proteins per cell line, respectively (Figure 1C). Intriguingly, some galactosyltransferases, carbohydrate sulfotransferases, collagens, and laminins were pulled down in both cell lines, suggesting consistent direct interactions with ECM proteins as well as intracellular metabolic enzymes that regulate glycoprotein synthesis. Further, this data indicates that there are interactions of CRP that may be conserved across epithelial-like cells as well as a certain degree of context-specificity in terms of CRP-bound complex formation dependent on cell type.

To expand on the recently identified PPIs for CRP through AP-MS, utilization of the BioGRID database [24,25] allowed for the inclusion of unique proteins identified from 11 other previously published earlier works that utilized various other techniques (e.g., yeast two-hybrid screening) of direct identification of PPIs [15,23,24,25,26,27,28,29,30,31,32,33,34,35]. This allowed for the expansion of the CRP interactome to 71 unique PPIs (Appendix A). This data was then imported into ShinyGO [36] for pathway analysis of gene set enrichment to identify pathways potentially regulated by CRP through direct interactions and to establish a revised interaction network for CRP. The investigation of the updated dataset of CRP PPIs identified significant enrichment of numerous pathways, including those involved in biosynthesis of glycosaminoglycans, extracellular matrix (ECM)–receptor interactions, the regulation of focal adhesions, advanced glycosylated end-product receptor signaling (AGE-RAGE), and various mitogenic and cancer-associated pathways (e.g., TGFβ, PI3K-Akt) (Figure 2). These pathways separated into two primary clusters, namely, those associated with (1) pathways in cancer and pathogen-induced inflammatory diseases, and (2) metabolic pathways and biosynthetic processes (Figure 3A), which is consistent with hierarchical clustering separating into these into two distinct categories (Figure 3B).

## 3. Potential Pathways Regulated by C-Reactive Protein

In-depth analysis using Gene Set Enrichment Analysis [37] via the Molecular Signature Database [38] provided further detail regarding which CRP PPIs were essential for the enriched pathways identified. Evaluation of the KEGG dataset [39] showed enrichment through laminins (LAMA1, LAMA5, LAMB1, LAMB2, LAMC1, LAMC3) that are associated with ECM–receptor interactions, focal adhesion, and pathways in cancer (Table 1). Biosynthesis of glycosaminoglycans (GAGs) heparan sulfate, keratan sulfate, and chondroitin sulfate, as well as biosynthesis of glycosphingolipids, were also significantly enriched due to the interaction of CRP with galactosyltransferases (B4GALT1, B4GALT3, B4GALT4, B4GALT7), and in the specific context of GAGs several sulfotransferases as well (HS3ST1, HS6ST1, NDST2, CHST3, CHST6, CHST12, CHST14), consistent with its potential interaction with various glycosyltransferase and other metabolic processes that regulate glycoprotein and glycolipid synthesis (Table 2 and Appendix A).

Additional analysis indicates several key pro-tumorigenic processes may also be altered by CRP PPIs as determined through the Hallmarks of Cancer dataset [40] (Table 3 and Table 4 and Appendix A). This included the aforementioned transferases’ involvement in metabolic flexibility associated with enhanced glycolytic activity as well as cellular responses to hypoxia, both of which are generally in favor of tumor development and progression [41,42]. Similarly, the interactions of CRP with laminins LAMA1 and LAMC1, collagen IVα2 (COL4A2), fibronectin (FN1), TGFβ1, QSOX, and TIMP3 significantly enriched the epithelial-to-mesenchymal transition (EMT) pathway, suggesting a potential direct impact of CRP on this process critically involved in malignant transformation and metastasis. Whether or not these particular biological activities contribute to the role of CRP in disease prevention and mitigation, or conversely, on the promotion of pathological processes, will require significant meticulous investigation going forward. Regardless, the updated CRP interactome indicates its potential involvement in metabolic processes, particularly those responsible for the structural modeling of the ECM and modulation of cell–cell interactions, that may have significant novel regulatory activities in inflammatory diseases and cancers that have yet to be described experimentally, as discussed in more detail in the following sections.

## 4. Possible Impact of C-Reactive Protein on Glycosylation of Proteins and Lipids in Cancer

### 4.1. Glycoproteins and the Role of Glycosaminoglycan Biosynthesis

Glycoproteins are proteins that have saccharide molecules attached to them and are critical in many biological processes. Located on the surface of cells or secreted into the extracellular matrix, glycoproteins help mediate cell signaling, cell adhesion, immune recognition, and tissue repair. In diseases like cancer, autoimmune disorders, and chronic inflammation, glycoprotein expression and glycosylation patterns can change, disrupting normal signaling and promoting pathological responses [43,44,45]. In cancer, alterations to glycosylation patterns and specific glycoprotein abundance have been mechanistically linked to signaling pathways involved in tumor cell proliferation, self-renewal, EMT, migration, invasion, immune invasion, and drug response (reviewed in [46,47]). Glycoprotein epitopes have been useful as biomarkers for diagnosis and risk stratification in patients in several cancers, and certain glycosylation patterns have been associated with resistance to chemotherapy [46,48,49]. Recent evidence by Rodriguez and colleagues has also suggested that glycosylation patterns, especially those in cancer cells within the tumoral compartment of the TME, are unique compared to healthy adjacent tissue [50]. It was also supported that specific glycosylation-related gene signatures were associated with patient outcomes in terms of survival and were observed to be cancer specific [50], highlighting the importance of glycobiology in cancer therapy. Further, multiple small molecule inhibitors targeting specific glycoproteins are currently under investigation clinically in multiple solid cancer types as well as leukemias, although the therapeutic benefit has yet to be determined (reviewed in [46]). Taken together, this data suggests that the distribution and composition of glycoproteins in the TME significantly impact tumor development, progression, and treatment resistance.

Glycosaminoglycans (GAGs) are long, negatively charged sugar chains that are typically attached to glycoproteins, forming proteoglycans. These structures are essential for supporting the physical and biochemical environment around cells. Typically, GAGs like heparan sulfate and keratan sulfate help organize signaling molecules, regulate inflammation, and control tissue hydration [44]. When dysregulated, as in diseases like Morquio A syndrome, defective GAG metabolism leads to tissue dysfunction, inflammation, and structural abnormalities [51]. Similar to what has been observed in terms of glycosylation patterns in cancer, alterations to GAG abundance and distribution have been associated with influencing tumor development and may be useful as diagnostic biomarkers in multiple cancer types (reviewed in [52]). Mechanistically, specific GAGs and the resulting glycoproteins have been shown to be involved in proliferation, adhesion, and invasion in cancer (as described further below).

Although limited data exists regarding CRP direct interactions with glycoproteins in the context of cancer, its ability to directly bind proteins in the ECM is well-established and continuously advancing. The direct binding of mCRP with several integral ECM glycoproteins, particularly in the endothelium of mice stimulated with IL-6, have been associated with the ability of mCRP to sustain endothelial activation and potentiate inflammation [18]. Furthermore, evidence indicates that mCRP-induced endothelial activation is amplified through the signaling downstream of FcyRIII, a low-affinity receptor for Fc fragments on antigen–IgG complexes that also serves as a CRP binding partner. Abnormalities in this multi-layered CRP-induced response might result in the overproduction of pro-inflammatory cytokines and lead to vascular dysfunction [53]. Considering the potential of CRP to regulate the biosynthesis of GAGs, such as heparan sulfate, keratan sulfate, and chondroitin sulfate (Table 1 and Table 2, Figure 2, Appendix A), it is possible that excessive CRP responses may modulate the production of key glycoproteins and exacerbate such dysfunction.

Together, GAGs, glycoproteins, and CRP form a dynamic and interconnected network during inflammation which adds a layer of complexity to the role of CRP in inflammation and the pathophysiology of diseases. Understanding this GAG–glycoprotein–CRP axis is crucial for improving diagnostics and developing targeted therapies. This is especially important considering that, while pCRP is used as a general inflammation marker, it does not fully reflect the more potent and localized activity of mCRP. Future tools that measure both isoforms and assess GAG or glycoprotein status may better guide treatment. Therapeutically, drugs that block the conversion of pCRP to mCRP or modulate how CRP interacts with GAG metabolism could reduce excessive inflammation without compromising necessary immune responses [54]. Overall, further understanding of how CRP influences GAG and glycoprotein synthesis may offer valuable insights into the biology of inflammation and disease. The novel CRP PPIs and the potential impact of CRP on specific GAG biosynthetic processes and glycoprotein production as it may relate to cancer is summarized below.

### 4.2. Heparan Sulfate

Heparan sulfate (HS) is a linear, sulfated glycosaminoglycan chain that is covalently attached to a protein core to form heparan sulfate proteoglycans (HSPGs). HSPGs play crucial roles in cellular processes by acting as co-receptors for a wide array of growth factors, morphogens, and ECM proteins. The impact of HS on HSPG function is governed by its unique structural features, including varying chain lengths and sulfation patterns that mediate specific electrostatic interactions with ligands. This structural variability is often dysregulated in cancer and regulates EMT, a critical step in cancer metastasis where epithelial cells gain mesenchymal traits and become more invasive [55,56].

Among the many ECM glycoproteins which interact with HS, FN1 plays a particularly critical role. It is a high-molecular-weight glycoprotein that is crucial for ECM structure and integrin-mediated cell adhesion, spreading, and migration. These interactions between FN1 and integrins have been shown to require HS-dependent syndecan-4 (SDC4) formation to facilitate the proper localization of FN1 on the cell surface for its participation in fibrillogenesis to regulate focal adhesions and cell motility [57]. HS-containing glycoproteins have also been evaluated as a potential therapeutic target, as the tetrabranched peptide NT4 specific to sulfated HS chains has shown potent inhibition of adhesion, migration, and colony formation of cancer cells [58]. This was in part owing to interference with SDC4-mediated signaling critical for regulating cell polarity and migration, consistent with the importance of HS in SDC4-FN1 coordinated cell motility [57]. Clinical trials investigating heparin as an anti-cancer therapeutic have been unsuccessful, despite some preclinical promise. Heparin appears to have inconsistent effects, such as increasing growth in some cell lines and reducing migration in others, depending on the tumor’s specific HSPG expression profile. Unlike NT4, which directly targets membrane-bound HSPGs and thereby modulates intracellular signaling and growth factor co-reception, soluble heparin often does not block HSPG function effectively in vivo, possibly due to redundancy in heparin-binding sites and the high concentrations needed for competition in the TME [58].

In light of the demonstrated ability of CRP to bind FN1 to potentiate an inflammatory phenotype in endothelial cells [18], and the importance of HS-dependent SDC4-FN1 interactions on nascent fibril formation in the ECM [57], the new observations that CRP may also regulate HS production suggest a potential additional activity in regulating cell adhesion and motility. Specifically, the direct interactions with B4GALT7, EXT1, and NDST2 could indicate that CRP plays a direct role in glycan chain initiation, elongation, and *N*-sulfation, respectively (Table 2 and Appendix A), that are required for HS conjugation to various peptides. However, it remains unclear whether these biosynthetic pathways are upregulated or downregulated in response to interaction with CRP, and therefore whether CRP may promote or limit HSPG formation and function.

### 4.3. Keratan Sulfate

Keratan sulfate (KS) is a complex and highly variable GAG composed of repeating disaccharide units of galactose and N-acetylglucosamine [59]. The structural diversity of KS arises from differences in chain length, sulfation patterns, and terminal saccharide modifications, which vary significantly between tissues. This variability plays an important role in tissue-specific functions and reflects the complexity of its biosynthetic pathway involving specific glycosyltransferases and sulfotransferases. KS modifies numerous proteoglycans (KSPG), particularly those in the extracellular matrix, and impacts processes like collagen fibrillogenesis, hydration, and the resilience of tissues [59]. A multitude of homeostatic mechanisms across numerous tissue types are subject to regulation by KSPGs (e.g., bone, brain, uterine, cervical, and corneal; reviewed in [60]).

Intriguingly, the abnormal accumulation of KS observed in Morquio A syndrome was shown to have an inverse correlation with plasma CRP [61]. Moreover, the use of a KS disaccharide L4 suppressed lipopolysaccharide-induced lung inflammation in mice in a model of chronic obstructive pulmonary disease [62], a disease in which CRP levels are generally significantly elevated [63]. In terms of cancer, the expression of KS-containing proteins has been reported in the tumor compartment of high-grade pancreatic cancer as well as astrocytoma [64,65]; however, how KS tumoral expression in these contexts relates to CRP levels was not defined. Some KSPGs, like mucin-1, are widely considered putative oncogenes (reviewed in [66]), whereas the KSPG lumican appears to be pro-tumorigenic in some cancers (e.g., enhancing proliferation, migration, and invasion) while instead having a protective effect in others [60,67,68,69]. The role of either is further complicated by their specific sulfation status [60]. Notably, early reports had identified the capacity for lumican to directly bind C1q of the complement 1 complex involved in the initiation of the classical pathway leading to complement cascade [70]. Given that CRP has been well-described to bind C1q and regulate compliment activation as well [18,71], the novel potential role of CRP in KSPG synthesis may increase the complexity by which CRP interacts with the tissue microenvironment to modulate local cell–cell signaling.

Taken together, this data suggests that the relationship between CRP and KSPGs is both context-dependent and disease-specific. The observed potential interaction of CRP with B4GALT3, B4GALT4, and CHST6 may contribute to the regulation of N- and O-linked glycosylation utilizing KSI and KSII (Table 2 and Appendix A). In light of this possibility, it will be critical to assess what correlative relationship CRP has with KS-containing proteins in the TME of various cancers. The potential for direct interactions warrants investigation on whether CRP may regulate biosynthesis and modification necessary for mucins, lumican, and other KSPGs in order to understand the impact of this interaction between CRP and KSPGs on disease development and progression.

### 4.4. Chondroitin Sulfate

Chondroitin sulfate (CS) is a sulfated glycosaminoglycan composed of repeating disaccharide units of N-acetylgalactosamine and glucuronic acid, which are variably sulfated at numerous positions. It is one of the most abundant GAGs in the human body and plays an essential role in the structural integrity and function of connective tissues, particularly cartilage. CS is a key part of the extracellular matrix, where it provides resistance to compression, modulates cell signaling, and helps support tissue hydration. It is naturally present in the cartilage, bone, skin, cornea, and arterial walls and contributes to tissue repair and regeneration. Clinically, CS has been extensively studied in the context of inflammatory diseases such as osteoarthritis as well as neuropathic pain and neurodegenerative diseases [72,73].

The influence of CS on inflammation and tissue repair is largely context-dependent. It has been shown to suppress inflammatory markers such as TNFα, NFκB, and CRP in animal models of neuropathic pain as well as to restore antioxidants (e.g., superoxide dismutase and catalase) to reduce oxidative stress and promote nerve tissue regeneration [73]. Conversely, CS proteoglycans (CSPG) were observed to potentiate inflammation through the suppression of inflammation resolution from spinal cord injury through modulating immune cell phenotypes, as enzymatic digestion of CSPGs promoted the differentiation of macrophages towards an M2 phenotype [74]. Moreover, the direct inhibition of CSPGs was also shown to promote tissue repair from demyelination injury through restoring astrocyte function [75].

In cancer, elevated CS levels have been associated with high CRP and metalloproteinase expression in the pleural effusions of patients with certain malignancies [76]. High expression of CSPGs, such as CSPG4, has also been reported in melanoma, prostate, and colon cancer and provides a potential therapeutic target for the delivery of cytotoxic small molecule payloads [77,78]. The elevation of another important CSPG, versican, has been observed in ovarian, lung, and breast cancer and may be associated with poorer outcomes [79,80,81,82]. Consistently, various CSPGs, in a sulfation-dependent manner, have been observed to regulate cell signaling pathways involved in metastatic processes such as cell adhesion, migration, and invasion in multiple cancer types in vitro and in vivo (reviewed in [83]), supporting a significant influence of CS-containing glycoproteins on tumor progression.

The recent AP-MS data indicates that CRP may potentially regulate CS synthesis through direct binding to B4GALT7, for the essential galactose conjugation to CS, as well as CHST3 and CHST12, which are responsible for the sulfation that modulates CSPG activity (Table 2 and Appendix A). Considering the potential dual role of CS in regulating both anti- and pro-inflammatory signaling, as well as the complex effects of CRP isoforms on pro-inflammatory and pro-resolution signaling (reviewed in [84]), it is possible that the effect of CRP on CS-related synthetic processes further mediates signaling within the microenvironment that influences cancer progression. However, it is currently unclear what direct effect CRP (of binding to B4GALT7, CHST3, etc.) may have on CS synthesis, how this alters CSPG activity, and thus whether this directly modifies tumor cell behavior. Understanding how these interactions with CRP influence CSPG function may elucidate their impact on cell signaling in the TME and their role in tumoral immune responses.

### 4.5. Glycosphingolipids

Glycosphingolipids are a class of amphipathic lipids that consist of a ceramide backbone linked to one or more sugar residues. These complex lipids are vital components of plasma membranes and are particularly enriched in lipid rafts, where they contribute to membrane microdomain organization, signal transduction, and cellular recognition. The dysregulation of glycosphingolipid metabolism is implicated in the onset and progression of various diseases, including cancer, neurodegenerative disorders, metabolic syndromes, and lysosomal storage diseases [85,86,87,88]. In cancer, ceramide typically promotes apoptosis, while S1P facilitates proliferation and angiogenesis, highlighting the opposing roles these lipids can play depending on the cellular context, especially considering that the relative levels of ceramide derivatives can significantly influence tumorigenicity [89,90]. Additionally, glycosphingolipids contribute to altered sensitivity to chemotherapy as well as influencing tumor–immune interactions (reviewed in [89,91]).

Sphingomyelin (SM) is an essential component of cell membranes that is involved in interactions between cells and the microenvironment and modulates cell–ECM interactions that influence many tumorigenic processes [91,92]. As SM is produced via phosphocholine conjugation to ceramide [90], and the capacity for CRP to bind phosphocholine-containing ligands has been well described (reviewed in [93]), the potential direct interaction between CRP and SM may possibly affect the signaling of either, but has not been directly assessed. The ability of CRP to directly bind B4GALT1, B4GALT3, B4GALT4, and ST3GAL4 may also impact the glycosylation of ceramide and its intermediates (specifically through galactosylation and sialylation) (Table 2 and Appendix A) to further influence the function and activities of ceramide, in addition to generally altering the sphingolipid rheostat, which itself could have broad implications on tumor development, progression, and treatment resistance. Moreover, as ceramide analogs have been shown to potentiate IL-1β/IL-6-dependent CRP production [94], it is possible that these novel additional interactions support reciprocal crosstalk between these molecules that provide another level of feedback. Further analysis of the relationship between CRP levels/activity and ceramide derivatives could provide insight on whether and how CRP could regulate the effects of ceramide/SM/S1P on tumor cells and their microenvironment.

## 5. Expanding on the Interactions of C-Reactive Protein with ECM–Receptor Signaling

ECM remodeling is considered one of the core processes that mediates a tumor cell’s ability to overcome otherwise impermissible conditions, allowing invasion into adjacent tissues, intravasation into systemic or lymphatic circulation, as well as adhesion and extravasation to distant tissues [95]. To this effect, fibronectin, collagens, and laminins appear to not only serve a structural purpose (e.g., fibrillogenesis, etc.), but interact directly with integrins and other proteins to regulate metastatic processes such as EMT [95,96,97,98,99,100] as well as immune evasion [96,97,101]. Prior in vitro studies indicate that CRP binds to laminin in a Ca^2+^-dependent manner via the phosphorylcholine binding site of CRP. This interaction was not observed to alter the cell attachment-promoting activity of laminin [14]. Further, while the mCRP domain associated with fibronectin and collagen binding was essential for its ability to induce IL-6 secretion from peritoneal immune cells (e.g., macrophages) [18], how this may reflect the outcome of its interactions with these glycoproteins in a tumor in situ has yet to be described. Although limited data specific to cancer exists, it was recently shown that CRP could directly bind integrin α2 and FcγRI in breast cancer to promote proliferation, adhesion, and invasion in cancer cells both in vitro and in vivo [10]. Taken together, this data suggests cell-dependent phenotypic responses to the direct interaction of CRP with ECM components and surface membrane receptors.

The regulation of ECM composition by mCRP has also been suggested, as it has been shown to upregulate cell adhesion molecules ICAM-1, VCAM-1, and E-selection in endothelial cells [102,103], P-selectin in platelets [104], and CD11b/CD18 in neutrophils [105], which was linked to altered activation states and cell adhesion. Similarly, while unattributed to a specific isoform, CRP has been indicated to induce metalloproteinase (MMP) expression in macrophages (MMP1 [106] and MMP9 [107]), endothelial cells (MMP1 and MMP10 [108]), and vascular smooth muscle (MMP2 [109] and MMP9 [110]), which suggests that CRP may alter the ability of cells to break down collagen and possibly other components of the extracellular matrix. The expression and activity of MMPs is, in part, mediated through ECM–receptor (e.g., integrin) interactions and is an essential step in ECM remodeling that regulates tumor cell invasion and extravasation [111]; however, the ability of MMPs to promote or suppress tumorigenic processes depends on a number of factors [111,112]. Whether CRP has similar upregulating effects on glycoprotein and MMP expression and activity in the context of cancer remains poorly understood; thus, the role of CRP in ECM remodeling in tumor cells as well as other cell types within the TME will require rigorous evaluation. Regardless, the PPIs of CRP with FN1, COL4A2, and multiple laminins (LAMA1, LAMA5, LAMB1, LAMB2, LAMC1, LAMC3) may suggest a potential regulation of integrin activity through one or more of these interactions (Table 2 and Appendix A). This behavior could in turn regulate numerous processes involved in tumor cell adhesion, migration, and invasion, although at present it is unclear if its activities in this context would be in favor of or opposed to tumor development and progression.

Lastly, it is also notable that, through the PPIs described here, CRP may also influence focal adhesion signaling pathways that promote cell migration and cytoskeletal rearrangement through its interactions with these ECM proteins, in addition to MAPK1 and MAPK3 (Table 1 and Appendix A), kinases generally associated with tissue remodeling that promote metastasis [113]. As such, CRP may alter cell–matrix adhesion and tissue architecture, as well as modulate cell–cell signaling that affects tumor progression. As eight of the twelve PPIs identified to enrich the “Pathways in Cancer” molecular signature are ECM glycoproteins, it is thus likely that some of the effects of CRP in the context of tumorigenicity specifically involve ECM–receptor signaling among various cell types between the tumoral and stromal compartments. Similarly, this was observed to be the case in terms of the “Epithelial-to-Mesenchymal Transition” pathway, which predominantly consisted of ECM proteins (FN1, COL4A2, LAMA1, LAMC1), in addition to a few key regulators of EMT, namely TGFβ1 (described further below) (Table 3 and Appendix A). This may be especially relevant considering the potential for pro- or anti-tumorigenic activity of CRP appears to be largely context-dependent and likely involves immune and other stromal cell signaling. Altogether, these possibilities merit evaluation of CRP–ECM–receptor signaling in organotypic models of the TME that can capture the complex dynamic interactions between cell types. Data assessing these interactions may provide a better understanding of the conditions in which CRP signaling regulates tumorigenic activity and immune responsivity.

## 6. Novel Insights into Other Cancer-Related Processes Potentially Modulated by C-Reactive Protein

### 6.1. TGFβ1 Signaling Pathway

Transforming growth factor β1 (TGFβ1) is a cytokine that has a key role in embryonic maturation, inflammatory responses, and maintaining homeostasis and regeneration in adult tissue [114]. In cancer, it has been indicated as a tumor suppressor through inducing apoptosis and cytostasis in early stages of tumor development. Conversely, TGFβ1 may also elicit tumor promotion in late-stage cancer cells with loss of its inhibitory functions that promotes tumor angiogenesis, metabolic flexibility, EMT, and evasion of drug-induced apoptosis [115,116,117]. TGFβ1 function is regulated through direct or indirect activation by integrins (e.g., αVβ6 and αVβ8), proteases (e.g., MMP-2), reactive oxygen species, and high acidity in the microenvironment [114]. While data is limited on a direct relationship between TGFβ1 and CRP in cancer signaling, they have been shown to have parallels with one another in atrial fibrillation, as a previous report indicated that CRP promotes inflammation and cell death via TGFβ1 in cardiac muscle cells in a dose-dependent manner [118].

The regulation of TGFβ1 by integrin αVβ6 and αVβ8 is of particular interest here, given the ability of fibronectin, collagens, and laminins to interact directly with these glycoproteins (Appendix A and [119,120]), suggesting that a complex network of interactions exists between CRP, FN1, COL4A2, LAMA1, LAMC1, and TGFβ1 (Table 1, Table 2 and Table 3) that could influence TGFβ1 signaling relevant to its role as either a tumor suppressor or promoter. Consistently, the capacity for FN1 fibrillogenesis has been shown to enhance TGFβ1 activity to drive EMT [98], and may also be influenced by CRP binding to either, or both, within the ECM, and may regulate this interaction to some extent. It is also important to note that the identification of the TGFβ1-CRP interaction was only captured by AP-MS in the HCT116 cells, but not pulled down in the HEK293T cell line, suggesting that this interaction may be either cell type- or cancer-specific. Further assessment of the interactions between CRP and TGFβ1 may further articulate their complex relationships with regulating tumorigenic processes.

### 6.2. Epithelial-to-Mesenchymal Transition

EMT has been widely studied for its potential role in cancer. While essential in embryonic development and wound healing, it is well-known to also be a critical process that allows tumor cells to elicit a spectrum of phenotypes required for proliferation, self-renewal, metabolic flexibility, invasion, anchorage-independent survival, and adhesion to metastatic sites [56]. In cancer, Type III EMT involves epithelial cells losing their ability to hold more rigid structures during this transition and resulting mesenchymal expression allows for migration of the cells, making this process a target for cancer cells to use for migration into circulation and metastasis. It should also be noted that mesenchymal–epithelial transition (MET) might be necessary in tumor progression, as transitioning cells back to epithelial phenotypes allows for cancer cells to adhere to and colonize tissue [121]. There is difficulty in experimentally confirming this proposed mechanism, but new approaches to phenotype expression evaluation and analysis of molecular targets in cancer cell lines may provide more concrete understanding. There is also a concern with EMT mechanisms lending themselves to treatment resistance, and the potential mechanisms behind this. Due to the fluid state of cell differentiation in EMT and MET, it is possible that the efficacy of drug therapy can vary depending on whether the cell is expressing mesenchymal or epithelial phenotypes, as this influences proteome composition, which may alter target availability or provide resistance mechanisms [122].

The role of CRP in tissue inflammation and repair is undeniably as important as it is vast. Although reviewed extensively elsewhere [12,84,93], the ability of CRP to regulate (1) fibroblast motility and differentiation, (2) immune cell recruitment, adhesion, and invasion, and (3) endothelial angiogenic behavior all relate to the many pathways that are exploited by tumor cells to maintain the plasticity required to engage in EMT or MET, as needed, to grow and metastasize successfully. As alluded to above, the CRP PPIs enriched in the “Hallmarks of Cancer: Epithelial-to-Mesenchymal Transition” (Table 3 and Appendix A) largely relate to glycoproteins (COL4A2, FN1, LAMA1, LAMC), signaling molecules (TGFB1), or enzymes (QSOX1, TIMP3) directly related to ECM structure and remodeling that may regulate processes underlying cellular plasticity and/or be a result of phenotypic changes. The novel interactions and potential new activities of CRP on glycoprotein biosynthesis and thus ECM composition, ECM-dependent cell–cell signaling, as well as TGFβ1 and MAPK signaling, altogether support the hypothesis that the role of CRP in the microenvironmental control of tumor cell EMT/MET is likely immensely underappreciated currently, despite growing evidence indicating its importance in the TME [6].

### 6.3. Aerobic Glycolysis

One of the fundamental features of tumor biology is metabolic reprogramming toward aerobic glycolysis, commonly referred to as the Warburg effect. This adaptation enhances glucose uptake and lactate production despite the presence of oxygen, providing biosynthetic precursors while establishing an acidic, immunosuppressive microenvironment that supports tumor progression [41]. Although direct evidence of CRP regulating glycolysis in tumor cells remains limited, several lines of evidence suggest that CRP contributes to glycolytic programming through immune and metabolic pathways relevant to the TME.

Mechanistically, CRP has been shown to influence glycolytic reprogramming in innate immune cells. Phosphocholine-bound CRP synergizes with Toll-like receptor ligands to drive inflammatory cytokine production in macrophages via FcγRI/IIa–Syk–PI3K–AKT2 signaling, a program dependent on enhanced glycolysis [123]. This metabolic shift not only supports cytokine release but also sustains fatty acid synthesis and trained immunity, suggesting that chronic CRP exposure may enforce a glycolysis-dependent inflammatory state within the tumor milieu [124]. Additional studies in ischemic cardiomyocytes revealed that CRP can recognize anaerobic glycolysis by binding to lysophosphatidylcholine-rich membranes, effectively sensing metabolic stress [125]. Whether this stress-sensing mechanism extends to glycolysis-driven tumor cell phenotypes remains unresolved, but highlights CRP as a potential metabolic sentinel.

Clinically, the combined evaluation of CRP and LDH has emerged as a powerful prognostic tool. Elevated levels of CRP and LDH pretreatment correlate with poor overall survival in cervical cancer [126], with tumor burden and unfavorable outcomes in pediatric Ewing sarcoma [127], and serve as independent prognostic factors in primary CNS lymphoma [128,129]. These findings reinforce the concept that CRP reflects systemic inflammation, while LDH captures glycolytic tumor metabolism, together identifying aggressive, glycolysis-driven disease states. Importantly, aerobic glycolysis may also promote therapeutic resistance in part by impairing T-cell effector functions [41,130], while high CRP, often driven by IL-6 signaling, contributes to diminished therapy response in certain contexts [6]. Thus, tumors characterized by concurrent elevations of CRP and LDH may represent a subset particularly refractory to standard therapies.

The integration of these findings with the updated CRP interactome further underscores the potential link between CRP and tumor cell metabolism. Proteomic analyses revealed that numerous CRP PPIs are enriched in pathways associated with glycosylation, suggestive of altered glucose utilization and carbohydrate metabolism; however, direct interactions with key glycolytic enzymes (e.g., hexokinase, pyruvate kinase, etc.) was notably absent (Table 4). Further investigation of whether CRP significantly alters glucose metabolism should utilize metabolomic approaches to determine if CRP activity influences aerobic glycolysis in tumor cells, as this may reveal whether CRP modulates metabolic reprogramming or if it instead influences biosynthetic processes in response to altered tumor cell metabolism.

### 6.4. Tumor Hypoxia

Hypoxia is a state of oxygen deficiency primarily used by cells in a controlled and adaptive response to enhance cell resiliency and preserve function; however, the signaling pathways involved are often exploited by cancer cells to promote angiogenesis, metabolic reprogramming, and other mechanisms that promote tumor development and progression [42]. The hypoxia-inducible factors HIF1α and HIF2α are transcription factors central to controlling gene transcription related to tumorigenic processes [131], and may be important pharmacotherapeutic targets in cancer [132]. HIF1α upregulates the vascular endothelial growth factor (VEGF) and platelet-derived growth factor (PDGF) to induce angiogenesis in the TME as well as inducing glucose transporters GLUT1 and GLUT3 to enhance tumor cell glycolytic metabolism [133]. Although participating in metabolic regulation to a lesser extent, HIF2α is generally more involved in the capacity of tumor cells to maintain stem cell-like behavior (e.g., self-renewal), the tumorigenic signaling involved in adhesion and invasion, and may confer resistance to many chemotherapeutics [134,135].

The role of CRP in hypoxia and HIF signaling is still insufficiently understood, despite the association of elevated levels in disease states marked by hypoxic or ischemic conditions such as heart failure, stroke, and chronic obstructive pulmonary disease, and may further indicate worse outcomes in these pathologies [136,137,138]. Although limited data exists, HIF1α has been shown to induce the expression of CRP and other pentraxins in tumor cells and neurons [139,140,141], while CRP has been shown to stabilize HIF1α expression in adipose-derived stem cells as well as induce angiogenic activity in endothelial cells [142,143]. Whether CRP interacts with HIF2α or alters its activity remains to be determined.

The recently identified CRP PPIs showed an enrichment of hypoxia signaling (Table 4). However, rather than being enriched due to canonical HIF-regulated genes well-described in cancer, the individual genes involved suggest that the relevance of this pathway is largely through the upregulation of sulfotransferases related to HSPGs (EXT1, HS3ST1, and NDST2). As hypoxia has been shown to promote HSPG synthesis and activity in several contexts [144,145,146], it is therefore possible that CRP may modulate the biosynthetic responses to HIF1α transcriptional activity and consequently alter the effect of HIFs on ECM remodeling. This may be especially important in the context of the tumor microenvironment as this may regulate cell–cell signaling driven by HIF1α and HIF2α that, in general, favor mechanisms that bolster tumor growth and metastasis.

## 7. Concluding Remarks

The recent proteomics-based evaluation of CRP PPIs has markedly expanded the CRP interactome and supports numerous novel regulatory roles of CRP across multiple processes critical to cell–cell signaling (Figure 4). These include its potential interactions in biosynthetic processes regulating glycoprotein abundance and function as well as ECM–receptor signaling that may relate to its biological activities in tissue homeostasis and inflammation. Further, these and other signaling mechanisms identified (e.g., TGFβ signaling, EMT) may also broaden the role of CRP in regulating processes critical to neoplastic development, metastasis, recurrence, and treatment resistance in cancer. Overall, this data contributes to the ever-evolving understanding of the complex nature of this integral protein and may provide new avenues of study to elucidate the involvement of CRP in pathophysiologic responses.

It is important to note that the BioPlex data used was in the context of embryonic epithelial-like cells (HEK293T) and colorectal tumor epithelial cells (HCT116). Thus, this study presents, for the first time, a comprehensive assessment of the biological activity of CRP directly on cells of an epithelial phenotype. The differential pulldown of PPIs, despite having ~50% overlap between the two cell lines, is suggestive of cell-dependent variability of how CRP may behave in certain contexts and will likely be markedly different across other types of cells within any given microenvironment (e.g., endothelial, immune, etc.). However, these differences between these two cell lines could potentially relate to the inherently complicated phenotype of the HEK293T cells that, while presenting with epithelial morphology under standard culture conditions, may not be highly representative of cells of an epithelial origin [147]. Regardless, as the overwhelming majority of experimental reports to date have focused on the functional impact of CRP on immune cells, among other stromal cell types within the TME (e.g., endothelial cells), this data demonstrates an urgent necessity to meticulously evaluate how CRP may also directly regulate the epithelial compartment within solid tumors. This possibility indeed complicates the already complex role that CRP plays in modulating cell–cell signaling between the tumoral and stromal compartments to influence tumor progression; however, it will be critical to better understand its direct action on malignant cells in order to fully appreciate its impact on tumor development, progression, and treatment response.

Although it cannot be explicitly concluded based on the nature of the data analyzed, it is more than likely that many of the interactions observed are mediated through the monomeric isoform of CRP (mCRP) following dissociation from its pentamer at the phospholipid bilayer. This notion is strongly supported by the fact that many of the protein targets identified in recent studies are indeed intracellular, in some cases typically found within specific cellular compartments (e.g., golgi, etc.), consistent with the possibility that CRP may be present in the cytosol and multiple organelles within cells (Appendix A). The data presented could therefore suggest that mCRP may play a significant direct role in glucose utilization and proteoglycan production that could further influence the underlying intracellular and cell–cell signaling events involved in inflammation and tumor progression. Additionally, given that the majority of PPIs assessed do not directly indicate the outcomes of these interactions on these proteins’ activities (i.e., increasing or decreasing activity), at present it is not possible or legitimate to assume the exact nature of how CRP is influencing these processes. However, especially in light of the significant breadth of clinical evidence suggesting that CRP is strongly associated with higher incidence and worse outcomes in many types of cancer [6], it is tempting to postulate that CRP may influence tumor development and progression in part through directly regulating proteins involved in biosynthetic processes, including for enhanced production of proteoglycans that mediate pro-tumorigenic cell–cell signaling. Regardless, as described elsewhere, it is clear that whether CRP has pro- or anti-tumorigenic activity is context-dependent and will be conditional based on many factors, including the microenviromental milieu. Similarly, as suggested above, many of the ECM-related enzymes and pathways identified in this work could in fact either support or suppress tumor development and progression through mediating cell–cell signaling and immune responsivity. These possibilities warrant systematic experimental investigation that utilize more direct approaches to evaluate the regulation of these signaling pathways through integrating cellular transcriptomic, epigenomic, and proteomic responses to CRP. This could be complemented by verifying the basal proteomic landscape of CRP PPIs across cancers and in stromal cell types. Altogether, this updated CRP interactome may provide a path forward to advance our understanding of this integral protein in regulating the pathophysiology underlying many disease states and may be especially useful in broadening our knowledge of how CRP modulates processes critical to tumorigenesis.

## Figures and Tables

**Figure 1 cimb-47-01003-f001:**
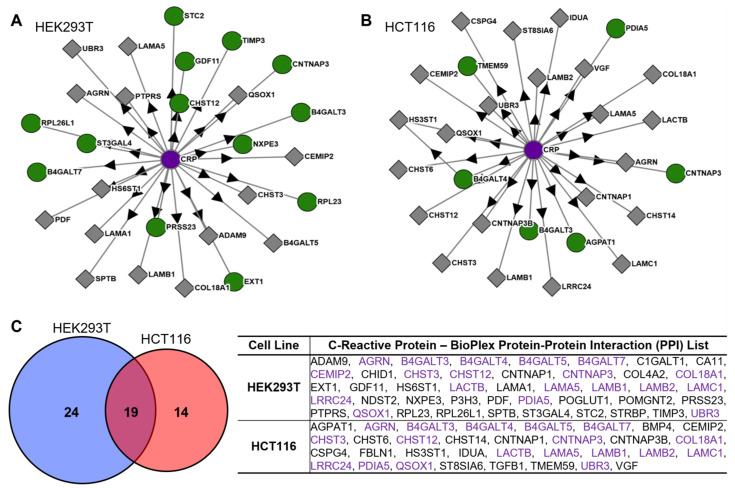
BioPlex Interactome of C-Reactive Protein. Protein–Protein interactions (PPIs) of C-reactive protein (CRP) in HEK293T (**A**) and HCT116 (**B**) cells identified by Huttlin et al., 2021 [23], using affinity pulldown mass-spectrometry (AP-MS). Purple node indicates target protein (CRP), with green nodes indicating target protein was both bait and prey and gray nodes indicating target protein was identified as prey; arrows indicate directionality of pulldown (i.e., anti-CRP immunoprecipitation). (**C**) Left: CRP PPIs identified by AP-MS in HEK293T and HCT116 included 24 unique proteins and 14 unique proteins, respectively, with 19 proteins identified in both, for a total of 57 PPIs altogether. Right: Protein–Protein interactions in both HEK293T and HCT116 cell lines, with overlapping proteins identified in purple (Overlap: AGRN, B4GALT3, B4GALT4, B4GALT5, B4GALT7, CEMIP2, CHST3, CHST12, CNTNAP3, COL18A1, LACTB, LAMA5, LAMB1, LAMB2, LAMC1, LRRC24, PDIA5, QSOX1, UBR3). Data was obtained using BioPlex Explorer 3.0 (https://bioplex.hms.harvard.edu/explorer/home, accessed on 1 May 2025).

**Figure 2 cimb-47-01003-f002:**
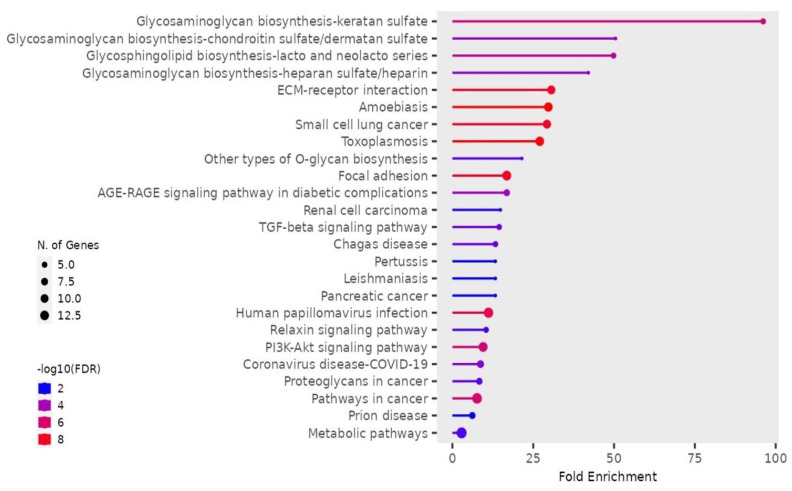
Visualization of KEGG Database Signaling Pathway Enrichment. Gene Set Enrichment Analysis (GSEA) was performed using the ShinyGO database (https://bioinformatics.sdstate.edu/go/, accessed on 1 May 2025). All 71 proteins with protein–protein interactions (PPIs) with CRP curated by BioGrid (https://thebiogrid.org) from published data were used and the KEGG Legacy (Kyoto Encyclopedia of Genes and Genomes) database was queried. Signaling pathways are listed by fold enrichment and ranked in descending order. Enrichment is determined based on the number of CRP PPIs input that overlap with the total number of proteins in each dataset/signaling pathway. Color indicates false-discovery rate (FDR) q-value, with red indicating highly significant (FDR q < 0.000000005) and blue indicating somewhat significant (FDR q < 0.005). Size of node indicates the number of genes, ranging from 5 to 12 CRP PPIs identified in the gene set. The top 5 most statistically significant enriched signaling pathways observed were involved in biosynthetic processes related to glycosaminoglycans (Keratan Sulfate, Chondroitin Sulfate, Heparan Sulfate) and glycosphingolipids as well as the “ECM–Receptor Interactions” dataset. The highest number of proteins associated with specific pathways included “ECM–Receptor Interactions”, “Pathways in Cancer”, “Metabolic Pathways”, “Focal Adhesions”, “Small Cell Lung Cancer”, “PI3K–Akt Signaling Pathway”, “HPV Infection”, “Amoebiasis”, and “Taxoplasmosis”.

**Figure 3 cimb-47-01003-f003:**
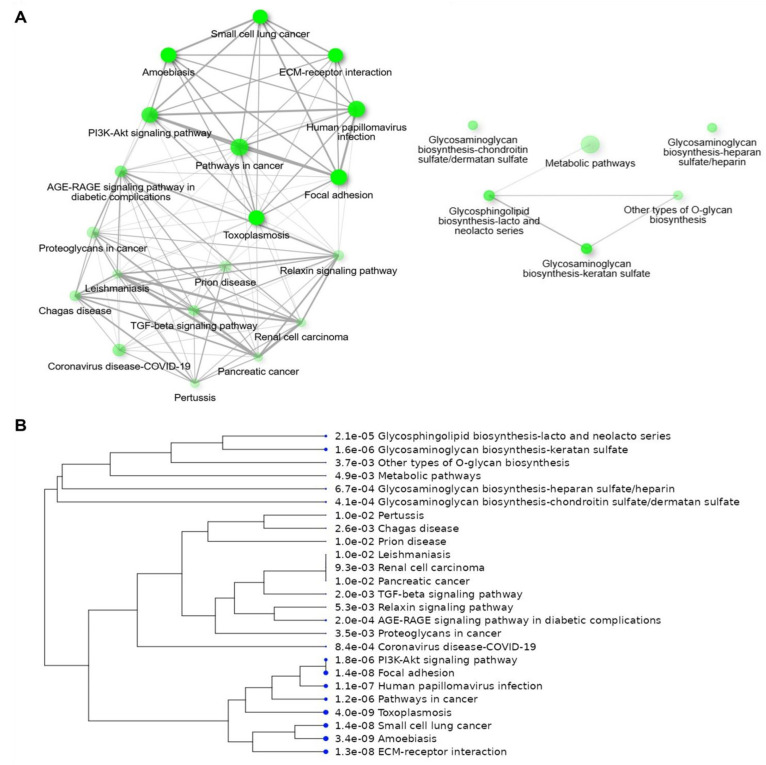
Cluster Analysis of Signaling Pathways Overrepresented in CRP PPIs. All 71 proteins with protein–protein interactions (PPIs) with CRP curated by BioGrid (https://thebiogrid.org) from published data were used and the ShinyGO database (https://bioinformatics.sdstate.edu/go/, accessed on 1 May 2025) database was queried. (**A**) Interactions between proteins and pathways identified by CRP PPIs showing similarities and overlap between specific pathways. The number of CRP PPIs identified in each pathway is indicated by node size with opacity indicating statistical enrichment. Number of proteins overlapping (i.e., CRP PPIs present in each dataset) is indicated by line thickness. (**B**) Hierarchical cluster analysis of pathways as a function of representation from the CRP PPI dataset, with PPIs shared across pathways clustering closer together. Data indicates two primary clusters: (top) metabolic pathways, specifically those associated with glycosaminoglycan and glycosphingolipid synthesis, and (bottom) cancer associated pathways and molecular responses to infectious diseases.

**Figure 4 cimb-47-01003-f004:**
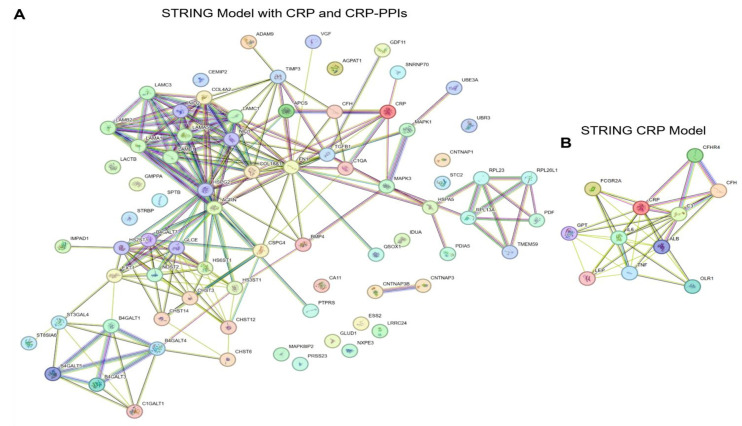
Updated Model of the CRP Interactome. STRING database (https://string-db.org/, accessed on 1 May 2025) analysis of the interaction network of all CRP PPIs showing known and predicted interactions between all proteins pulled down by CRP (**A**) compared to the general inquiry of the STRING database only for CRP (**B**). Line colors and indication: cyan (database curation), magenta (experimentally determined), green (gene neighborhood), red (gene fusion), blue (gene co-occurrence), yellow (text-mining), black (co-expression), and light blue (protein homology). Absence of a line indicates that the specific protein has not been documented in the STRING database to interact with the other proteins identified as CRP PPIs. Querying the STRING database utilizing the updated CRP interactome provides numerous additional signaling networks and indicates their interrelatedness based on CRP PPIs (**A**), suggesting the potential broader impact of CRP on cell signaling in comparison to its traditional CRP STRING model based on traditionally described interactions (**B**).

**Table 1 cimb-47-01003-t001:** Potential Regulation of Metabolic Processes and Tumorigenic Signaling by C-Reactive Protein. Gene Set Enrichment Analysis (GSEA) was performed using the Molecular Signature Database (https://gsea-msigdb.org, accessed on 1 May 2025). All 71 proteins with protein–protein interactions (PPIs) curated by BioGrid (https://thebiogrid.org, accessed on 1 May 2025) from published data were used and the KEGG Legacy (Kyoto Encyclopedia of Genes and Genomes) database was queried. Signaling pathways are indicated with the number of proteins identified per pathway, with the false-discovery rate (FDR) q-value and the specific PPIs per signaling pathway identified. # indicates number of genes present in overlap.

KEGG Legacy Pathways	# Genes in Overlap (k)	FDR q-Value	CRP PPIs
ECM–Receptor Interaction	9	3.65 × 10^−12^	AGRN, COL4A2, FN1, LAMA1, LAMA5, LAMB1, LAMB2, LAMC1, LAMC3
Pathways in Cancer	12	2.54 × 10^−11^	BMP4, COL4A2, FN1, LAMA1, LAMA5, LAMB1, LAMB2, LAMC1, LAMC3, MAPK1, MAPK3, TGFB1
Focal Adhesion	10	7.29 × 10^−11^	COL4A2, FN1, LAMA1, LAMA5, LAMB1, LAMB2, LAMC1, LAMC3, MAPK1, MAPK3
Small-Cell Lung Cancer	8	7.29 × 10^−11^	COL4A2, FN1, LAMA5, LAMB1, LAMB2, LAMC1, LAMC3, LAMA1
GAG Biosynthesis—Heparan Sulfate	5	2.66 × 10^−8^	B4GALT7, EXT1, HS3ST1, HS6ST1, NDST2
Prion Diseases	5	1.08 × 10^−7^	C1QA, HSPA5, LAMC1, MAPK1, MAPK3
GAG Biosynthesis—Keratan Sulfate	4	2.54 × 10^−7^	B4GALT1, B4GALT3, B4GALT4, CHST6
GAG Biosynthesis—Chondroitin Sulfate	4	1.18 × 10^−6^	B4GALT7, CHST3, CHST12, CHST14
Glycosphingolipid Biosynthesis	4	2.13 × 10^−6^	B4GALT1, B4GALT3, B4GALT4, ST3GAL4
TGFβ Signaling Pathway	4	2.53 × 10^−4^	BMP4, MAPK1, MAPK3, TGFB1

**Table 2 cimb-47-01003-t002:** Protein–Protein Interactions of C-Reactive Protein Involved in Glycoprotein Synthesis and ECM–Receptor Signaling. Gene Set Enrichment Analysis (GSEA) was performed using the Molecular Signature Database (https://gsea-msigdb.org, accessed on 1 May 2025). All 71 proteins with protein–protein interactions (PPIs) curated by BioGrid (https://thebiogrid.org, accessed on 1 May 2025) from published data were used and the KEGG Legacy database was queried. Annotations of the CRP PPIs identified in the KEGG pathways associated with Glycosaminoglycan and Glycosphingolipid Synthesis (top) and ECM–receptor Signaling (bottom) are provided, with a summary of each enzyme/protein as indicated by NCBI (https://www.ncbi.nlm.nih.gov) and Uniprot (https://www.uniprot.org) databases accessed through GeneCards (https://www.genecards.org).

Signaling Pathway	Gene	Function
**Glycosaminoglycan and**	B4GALT1	Transfer of galactose to GlcNAc residues in glycoproteins, required for saccharide structuring and initiation for glycosaminoglycan synthesis
**Glycosphingolipid**	B4GALT3	
**Biosynthesis**	B4GALT4	
	CHST3	Catalyzes sulfation of Chondroitin Sulfate GalNAc residues at position 6
	CHST6	Catalyzes sulfation of Keratan Sulfate GlcNAc residues
	CHST12	Catalyzes sulfation of Chondroitin Sulfate GalNAc residues
	CHST14	Catalyzes sulfation of Dermatan Sulfate GalNAc residues
	EXT1	Required for chain elongation of Heparan Sulfate glycan backbone
	HS3ST1	Catalyzes sulfation of Heparan Sulfate GlcNAc residues at position 3; rate limiting
	HS6ST1	Catalyzes sulfation of Heparan Sulfate GlcNS residues at position 6
	NDST2	Catalyzes N-deacetylation and N-sulfation of Heparan Sulfate GlcNAc
	ST3GAL4	Catalyzes sialylation of glycoproteins and glycolipids important for cell adhesion
**ECM–Receptor Signaling**	AGRN	Heparan Sulfate glycoprotein involved in cholinergic receptor and cell–cell signaling
	COL4A2	Major structural component of ECM making up the basement membrane
	FN1	Binds various ECM glycoproteins to regulate cell cytoskeleton, adhesion, and motility
	LAMA1	Subunits of laminins (alpha, beta, gamma) Major glycoproteins of ECM involved in cell adhesion and motility Directly interact with multiple types of integrins for cell–cell signaling May directly bind glycosphingolipids to alter ceramide-derivative signaling Implicated in EMT and metastatic processes
	LAMA5	
	LAMB1	
	LAMB2	
	LAMC1	
	LAMC3	

**Table 3 cimb-47-01003-t003:** Protein–Protein Interactions of C-Reactive Protein Involved in TGFβ Signaling and Epithelial-to-Mesenchymal Transition. All 71 proteins with protein–protein interactions (PPIs) curated by BioGrid (thebiogrid.org, accessed on 5 January 2025) from published data were used and the KEGG Legacy and Hallmark databases were queried. Annotations of the CRP PPIs identified in the KEGG pathways associated with TGFβ Signaling Pathway (top) and in the Hallmarks of Cancer pathways associated with epithelial-to-mesenchymal transition (bottom) are provided, with a summary of each enzyme/protein as indicated by NCBI (ncbi.nlm.nih.gov) and Uniprot (uniprot.org) databases accessed through GeneCards (genecards.org, accessed on 5 January 2025).

Signaling Pathway	Gene	Function
**TGFβ Signaling Pathway**	BMP4	Ligand for TGFβ receptors that regulates SMAD signaling
	MAPK1	ERK2 kinase involved in cell proliferation, adhesion, motility, and survival
	MAPK3	ERK1 kinase involved in cell proliferation, adhesion, motility, and survival
	TGFB1	Major growth factor involved in MAPK and SMAD signaling associated with cell proliferation, differentiation, adhesion, motility; regulates immune cell function
**Epithelial–Mesenchymal Transition**	COL4A2	Major structural component of ECM making up the basement membrane
	FN1	Binds various ECM glycoproteins to regulate cell cytoskeleton, adhesion, and motility
	LAMA1	Alpha and gamma subunits of laminin involved in cell–cell signaling with integrins associated with cell proliferation, differentiation, adhesion, and motility
	LAMC1	
	QSOX	Required for the incorporation of laminins into the extracellular matrix
	TGFB1	Major growth factor involved in MAPK and SMAD signaling associated with cell proliferation, differentiation, adhesion, motility; regulates immune cell function
	TIMP3	Inhibits matrix metalloproteinases (e.g., MMP2) to prevent ECM degradation; activity is associated with tumor suppression

**Table 4 cimb-47-01003-t004:** Protein–Protein Interactions of C-Reactive Protein Involved in Glycolysis and Hypoxia. All 71 proteins with protein–protein interactions (PPIs) curated by BioGrid (thebiogrid.org, accessed on 5 January 2025) from published data were used and the KEGG Legacy and Hallmark databases were queried. Annotations of the CRP PPIs identified in the Hallmarks of Cancer pathways associated with Glycolysis (top) and Hypoxia (bottom) are provided, with a summary of each enzyme/protein as indicated by NCBI (ncbi.nlm.nih.gov) and Uniprot (uniprot.org) databases accessed through GeneCards (genecards.org, accessed on 5 January 2025).

Signaling Pathway	Gene	Function
**Glycolysis**	AGRN	Heparan Sulfate glycoprotein involved in cholinergic receptor and cell–cell signaling
	B4GALT1	Transfer of galactose to GlcNAc residues in glycoproteins, required for saccharide structuring and initiation for glycosaminoglycan synthesis
	B4GALT4	
	B4GALT7	
	CHST6	Catalyzes sulfation of Keratan Sulfate GlcNAc residues
	CHST12	Catalyzes sulfation of Chondroitin Sulfate GalNAc residues
	EXT1	Required for chain elongation of Heparan Sulfate glycan backbone
	HSPA5	Chaperone involved in protein folding and assembly; may regulate unfolded protein response
	GMPPA	Synthesis of GDP-mannose for N-linked oligosaccharides
	IDUA	Catalyzes hydrolysis of Heparan Sulfate and Dermatan Sulfate
	QSOX1	Required for the incorporation of laminins into the extracellular matrix
	STC2	Involved in cellular calcium and phosphate homeostasis
**Hypoxia**	CHST3	Catalyzes sulfation of Chondroitin Sulfate GalNAc residues at position 6
	HS3ST1	Catalyzes sulfation of Heparan Sulfate GlcNAc residues at position 3; rate limiting
	HSP5	Chaperone involved in protein folding and assembly; may regulate unfolded protein response
	EXT1	Required for chain elongation of Heparan Sulfate glycan backbone
	NDST2	Catalyzes N-deacetylation and N-sulfation of Heparan Sulfate GlcNAc
	STC2	Involved in cellular calcium and phosphate homeostasis

## Data Availability

The original contributions presented in this study are included in the article/Appendix A. Further enquiries can be directed to the corresponding author. All data analyzed in this study was through the following bioinformatics databases, accessed in May 2025: BioGRID (https://thebiogrid.org/) was used to identify CRP protein–protein interactions as curated through that database. BioPlex Explorer 3.0 (https://bioplex.hms.Harvard.edu/explorer/home) was used to visualize the interactome data of protein interactions in HEK293T and HCT116 cells reported by Huttlin et al., 2021 [23]. The Molecular Signature Database (MSigDB, Broad Institute, https://www.gsea-msigdb.org/gsea/) was used to assess the genes identified from the BioGRID dataset for pathway analysis using the Gene Set Enrichment Analysis (GSEA) tool. The ShinyGO database (Version 0.82; https://bioinformatics.sdstate.edu/go/) was used to query the protein interactions to assess KEGG network pathways to visualize pathway enrichment. The STRING database (https://string-db.org/) was used to query interactions among proteins pulled down by CRP. The COMPARMTMENTS database (https://compartments.jensenlab.org) was used to evaluate known and predicted intracellular localization of CRP. All databases were accessed on 1 May 2025.

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
