# Peer review of "Evaluating the CRP Interactome: Insights into Possible Novel Roles in Cellular Signaling and Tumorigenicity"

_cimb, 2025, doi:10.3390/cimb47121003_

Round 1

Reviewer 1 Report

Comments and Suggestions for Authors

The manuscript presents a comprehensive & well-integrated review of the CRP interactome, consolidating both past findings as well as novel proteomics-based insights. Moreover, the paper effectively links molecular interactions of CRP with cellular signaling processes relevant to inflammation, extracellular matrix (ECM) regulation and tumor biology. This represents an important step forward in understanding the pleiotropic nature of CRP in disease. However, I would like add some suggestions to improve the paper further. Although presented as a “review,” portions of the manuscript such as Figures 1-3 resemble original bioinformatics analyses. It would strengthen the paper to clarify whether these analyses are new work or re-interpretations of existing datasets to avoid ambiguity about the study type. While the connections between CRP and ECM/glycosylation are well-explained, the link between CRP isoforms (pCRP, mCRP) and specific interactome findings could be more explicitly discussed. A short subsection contrasting how isoform localization might affect specific protein-protein interactions such as FN1, LAMA1, or B4GALT family enzymes would be valuable. Also, please double check grammar, language, and possible typo corrections such as “Affinty Capture” etc.

Comments on the Quality of English Language

Please check the comments above. 

Author Response

Reviewer 1 
The manuscript presents a comprehensive & well-integrated review of the CRP interactome, 
consolidating both past findings as well as novel proteomics-based insights. Moreover, the 
paper effectively links molecular interactions of CRP with cellular signaling processes relevant 
to inflammation, extracellular matrix (ECM) regulation and tumor biology. This represents an 
important step forward in understanding the pleiotropic nature of CRP in disease. However, I 
would like add some suggestions to improve the paper further.  

1. Although presented as a “review,” portions of the manuscript such as Figures 1-3 resemble 
original bioinformatics analyses. It would strengthen the paper to clarify whether these 
analyses are new work or re-interpretations of existing datasets to avoid ambiguity about 
the study type.  

The Authors thank the Reviewer for pointing out this discrepancy. The analysis provided in 
Figures 1-3 is using a combination of bioinformatics tools with curated databases of existing 
data, rather than a re-interpretation of data. While Figures 1-3 themselves are original, the data 
used to generate them is not, and the presented work is extensively a discussion reviewing 
relevance of this data in the context of the literature. For these reasons it did not seem 
appropriate for us to classify this manuscript as a primary research article but rather as a review 
article. We have revised the discussion of this data in Section 2 to more clearly articulate how 
this data was obtained and analyzed (highlighted portion at Lines 70-75 and Lines 102-105). We 
also included additional information in the figure legends for Figures 1-3 to improve clarity.

2. While the connections between CRP and ECM/glycosylation are well-explained, the link 
between CRP isoforms (pCRP, mCRP) and specific interactome findings could be more 
explicitly discussed. A short subsection contrasting how isoform localization might affect 
specific protein-protein interactions such as FN1, LAMA1, or B4GALT family enzymes would 
be valuable.  

We agree that this is a fascinating and exciting prospect to complement the current work. We 
have performed analyses using COMPARTMENTS and have found that there are some 
intriguing data regarding which CRP isoform could be interacting with these proteins based on 
localization. This data is included at the end of this cover letter (last page of pdf). Because some of the 
data curated by COMPARTMENTS is possibly derived from text-mining as the only source, a 
thorough analysis will be required to accurately represent the data and legitimately argue any 
claims of specific CRP isoforms in these interactions. We believe that further investigation of this 
through deliberate analysis of primary literature, UniProt, and other sources may justify a more 
in-depth evaluation of this possibility. Given the mounting evidence suggesting the importance 
of CRP isoforms in pathophysiology, this work may prove to be quite impactful and we believe 
that it will constitute a full manuscript on its own, especially as this may be more broadly 
applicable to numerous pathologies apart from cancer. The Authors sincerely thank the Reviewer 
for this great idea and are invested in pursuing this in subsequent work.

3. Also, please double check grammar, language, and possible typo corrections such as 
“Affinty Capture” etc. 

The Authors appreciate the Reviewer’s observation regarding typos, especially those that were 
in the Supplementary data file as indicated by the comment. To address this, we have extensively 
revised Supplementary Table 1 and have gone through the manuscript thoroughly to correct 
typos and errors in grammar or logic. We have also revised much of the manuscript, reducing 
sentence structure where possible, revising sentences for clarity, and removing numerous 
sentences to benefit the readability without altering the content. These changes were extensive 
throughout the manuscript and associated files. Significant changes to sentence structure/etc. 
were highlighted, but to maintain ease of review the typos and minor grammatical revisions were 
not highlighted. 

Reviewer 2 Report

Comments and Suggestions for Authors

In this review article, Gerhardt et al., describe C-reactive protein (CRP) as a key biomarker and regulator of inflammation. They highlight CRP’s diverse biological roles across its isoforms in initiating, amplifying, and resolving inflammation. The study emphasizes CRP’s influence on intercellular communication among various cell types within tissues. Finally, the authors propose that CRP may have novel regulatory roles in metabolism, extracellular matrix (ECM) dynamics, and tumor microenvironment signaling.

Topic #1 is interestig and marks the point where the authors begin to discuss the main subject of the work. In my opinion, it could be better structured. The authors may improve this section by streamlining background details, clarifying the transition from CRP’s clinical relevance to its molecular mechanisms, reducing redundancy about isoforms and disease associations, emphasizing the study’s specific research gap and objective earlier, and improving sentence structure for clarity and flow.

The captions for Figures 1, 2, and 3 should be rewritten. The figures are complex and contain a lot of information, but in my opinion, they are poor in content.

Regarding the topic #3, in  m,y opinion, it should be expanded. It addresses a broad subject but is presented with limited content.   Regarding the topic "Possible Impact of C-Reactive Protein on Glycosylation of Proteins and Lipids", before discussing glycosylation itself, the authors should frst address the importance of glycosylation under physiological and pathological conditions. This is a rapidly growing field of research, yet it remains somewhat neglected in the biomedical area.   Regarding Topic #5, the authors may improve it by organizing the discussion more coherently around key mechanisms, providing a concise summary of known CRP–ECM interactions before presenting new findings, and concluding with a clearer statement on how these interactions advance the understanding of CRP’s role in tumor progression. This could make the topic more fluid and facilitate understanding for readers, especially those who are not familiar with the subject addressed in the review.   The caption for Figure 4 should also be rewritten, as it is poor in content and confusing.            

Author Response

In this review article, Gerhardt et al., describe C-reactive protein (CRP) as a key biomarker and 
regulator of inflammation. They highlight CRP’s diverse biological roles across its isoforms in 
initiating, amplifying, and resolving inflammation. The study emphasizes CRP’s influence on 
intercellular communication among various cell types within tissues. Finally, the authors 
propose that CRP may have novel regulatory roles in metabolism, extracellular matrix (ECM) 
dynamics, and tumor microenvironment signaling. 

1. Topic #1 is interestig and marks the point where the authors begin to discuss the main 
subject of the work. In my opinion, it could be better structured. The authors may improve 
this section by streamlining background details, clarifying the transition from CRP’s clinical 
relevance to its molecular mechanisms, reducing redundancy about isoforms and disease 
associations, emphasizing the study’s specific research gap and objective earlier, and 
improving sentence structure for clarity and flow.

The Authors agree with the Reviewer that the Introduction could be improved through 
streamlining. Accordingly, we have entirely revised the Introduction to convey the most pertinent 
background to provide context for the review (Lines 35-60). We have also revised sentence 
structure throughout the manuscript to make the discussions of the data more concise and 
generally improve readability.

2. The captions for Figures 1, 2, and 3 should be rewritten. The figures are complex and contain 
a lot of information, but in my opinion, they are poor in content.

The Authors thank the Reviewer for this critique regarding the presentation of this data. To 
improve accessibility of this data to the reader, we have significantly revised the figure legends 
for Figures 1-4 to give more context. We have included additional description while limiting 
commentary to help the audience interpret the data effectively. We highly value this comment, 
as one of the goals of this manuscript is to provide this data in a meaningful way and as a usable 
reference to the field to assist in advancing experimental evaluation of these potential 
interactions and pathways.  

3. Regarding the topic #3, in  m,y opinion, it should be expanded. It addresses a broad subject 
but is presented with limited content.    

We appreciate the Reviewer’s concern regarding Section 3. Both Sections 2 and 3 are intended 
as presentation of the bioinformatics data with interpretation to establish a review of the literature 
in the most significant and novel pathways identified by the protein-protein interactions with CRP. 
This allowed us to further expand into the relevance of these pathways to cancer, the 
conventional understanding of CRP’s activities in terms of these pathways, and to interpret the 
novel data shared (from Sections 2 and 3) in these contexts. This was noted on Line 190 to 
clarify intent.

4. Regarding the topic "Possible Impact of C-Reactive Protein on Glycosylation of Proteins and 
Lipids", before discussing glycosylation itself, the authors should frst address the 
importance of glycosylation under physiological and pathological conditions. This is a 
rapidly growing field of research, yet it remains somewhat neglected in the biomedical area.

The Authors agree that this is an exciting and important field, and that there was a shortcoming 
in terms of providing adequate background to establish context. We have extensively revised 
this portion of the manuscript (Lines 199-215), providing pertinent information of glycobiology in 
cancer and have also provided numerous references of more comprehensive reviews on this 
topic to assist the reader. We believe that this new introduction to Section 4 will help provide 
better context to relate the CRP protein interaction data to its possible regulatory activity of   
glycosaminoglycan synthesis and glycosylation as it pertains to cancer.

5. Regarding Topic #5, the authors may improve it by organizing the discussion more 
coherently around key mechanisms, providing a concise summary of known CRP–ECM 
interactions before presenting new findings, and concluding with a clearer statement on how 
these interactions advance the understanding of CRP’s role in tumor progression. This could 
make the topic more fluid and facilitate understanding for readers, especially those who are 
not familiar with the subject addressed in the review.  

The Authors appreciate this comment from the Reviewer. In prior works we have reviewed this 
topic to some extent but we entirely agree that more context is needed for this particular work in 
order to benefit the reader. We have reorganized this section significantly and have provided 
additional information regarding known direct and indirect activities of CRP in ECM remodeling, 
which included an additional 14 references to provide proper background. These changes occur 
in the highlighted portion of Section 5, with major new content on Lines 422-448.

6. The caption for Figure 4 should also be rewritten, as it is poor in content and confusing.

In line with the response for Comment #2, we agree and have added additional description to 
help with the interpretation of this data. We believe that Figure 4 highlights the overall importance 
and utility of the manuscript as well as the approach we have taken using bioinformatics to inform 
our review of the literature to present novel areas of research on CRP.
